# Differences in Soil Water Holding Capacity and Available Soil Water along Growing Cycle Can Explain Differences in Vigour, Yield, and Quality of Must and Wine in the DOCa Rioja

José María Martínez-Vidaurre [1], Eva Pilar Pérez-Álvarez [1], Enrique García-Escudero [1], María Concepción Ramos [2] and Fernando Peregrina [3,*]

1   Instituto de Ciencias de la Vid y del Vino-ICVV, Universidad de La Rioja, Gobierno de La Rioja, CSIC, 26007 Logroño, Spain; jmvidaurre@larioja.org (J.M.M.-V.); evapilar.perez@icvv.es (E.P.P.-Á.); egescudero@larioja.org (E.G.-E.)
2   Departamento de Química, Física y Ciencias Ambientales y del Suelo, Universitat de Lleida-Agrotecnio Center CERCA, 25198 Lleida, Spain; mariaconcepcion.ramos@udl.cat
3   Área de Edafología, Departamento Producción Agraria, E. T. S. I. Agronómica, Alimentaria y de Biosistemas, Universidad Politécnica de Madrid, 28040 Madrid, Spain
*   Correspondence: fernando.peregrina@upm.es; Tel.: +34-910671046

**Abstract:** Soil water availability during the vine growth cycle can affect yield and grape quality. The objective was to evaluate the effect of soil water holding capacity (AWC) and available soil water (ASW) throughout the growing cycle on the nutritional status, vigor, production, and composition of grapes and wine. The study was conducted in the municipality of Uruñuela in the DOCa Rioja (Spain). The soils of four rainfed vineyard plots were characterized to determine AWC and its impacts on vine, grape, and wine composition. The N, P, and K foliar content, vigor, grape yield, berry weight, and composition of must and wine were analyzed in those vineyard plots during the period 2010–2014. The ASW was simulated in each plot and each year analyzed, considering the soil properties and the weather conditions, after model calibration in one plot in which soil water content was registered. The results showed that AWC influenced ASW along the growing cycle, so vines suffered from water stress in some periods of the vegetative cycle. Plots with higher AWC had higher ASW from fruit set to ripening and lower water stress during this period, which explains the higher N, P, and K foliar content, vigor and grape yield, and lower polyphenol and anthocyanin content in grapes and wines. The period where water availability had the most influence on the quality of the grapes was from veraison to ripening, during which ASW increased berry weight and acidity and decreased anthocyanins and polyphenolic compounds.

**Keywords:** soil rooting depth; polyphenols; anthocyanins; hydric stress

## 1. Introduction

Within the same climatic zone, the soil characteristics determine vine development and grape composition [1,2]. The depth of the soil, its texture, and porosity regulate the amount of soil accessible to roots, affecting the movement of water and nutrients, as well as soil water storage [3–5]. Furthermore, they impact water availability, subsequently influencing the water status of the vine [6,7]. Therefore, the water availability in the vineyard soil could be used to estimate the water stress on grapevines.

Water availability can affect the uptake by the vine of principal nutrients such as N, P, and K from the soil. Because the ionic forms $NO_3^-$, $H_2PO_4^-$, and $K^+$ are actively absorbed and require an energy input in the form of ATP, under hydric stress, the vine reduces its photosynthetic activity as a result of the stomata closure [8]. The reduction in the assimilation of these nutrients has a great impact on the vineyard, especially N, due to its great influence on vine vigor [9] and on the polyphenol and anthocyanin contents of the

grape [10]. While K affects the flow of water in the plant and the acidity conditions of the grape, which are crucial for obtaining a quality wine [11], water stress can affect grapevines at different times during the growth cycle. Additionally, shoot growth is reduced under any level of water stress [8]. Furthermore, deficits in water availability, occurring during both the early and late stages of the growing season, lead to a reduction in yields [12].

Berry composition is also affected by soil properties, vine management, cultivar, and climate [13]. Thus, water deficits of up to 50% of crop evapotranspiration (ETc) have a minimal effect on yield but yield decreases if this threshold is exceeded [14]. So, Ramos and Martínez-Casasnovas [15] found that this effect on yield occurred in years when the water deficit was higher than 50% ETc during the growing season.

Berry growth is affected by water deficit, which also has an impact on grape quality [16–20]. This impact depends on the intensity of water stress and also on when this stress appears [21–23]. Sugar accumulation in berries is affected by a late water deficit, as this water stress reduces leaf photosynthesis [24]. The increase in irrigation rates provokes an increase in total acidity, while the malic acid concentration decreases under both regulated irrigation and prolonged deficit irrigation, resulting in an increase in the tartaric/malic acid ratio [25]. In addition, water stress before veraison negatively affects titratable acidity, malic acid, and aromatic quality [26]. But this hydric stress can have a positive effect by increasing the concentration of anthocyanin and polyphenols [27].

Thus, in irrigation treatments, the total polyphenol content decreased, with the lowest anthocyanin and phenol concentrations under the irrigation rate corresponding to 100% of crop evapotranspiration (ETc) [28].

According to these results, Cooley et al. [25] and Lizama et al. [29] observed that wine color density, anthocyanin, and polyphenolic levels decrease with increasing irrigation rates [30]. Bucchetti et al. [30] reported that water deficits increase the anthocyanin content per berry, reduce fruit growth, and consistently increase anthocyanin concentration.

In areas where vines are usually cultivated under rainfed conditions because of water scarcity, the vine response mainly depends on soil water content, which will be driven by the rainfall amount, its distribution along the cycle, and the soil properties. Thus, high variability can be recorded from year to year.

This research aims to analyze the variability in the response of grapevines cultivated in rainfed conditions to various water stress scenarios and their impact on plant development. The research was conducted in Rioja DOCa on the Tempranillo (*Vitis vinifera* L.) cultivar, which, due to this appellation, is the largest producer of that variety worldwide. The aim was to analyze the variability in the response of grapevine to water stress under rain-fed conditions over five years (2010–2014). Focused on: (i) nutritional status, vigor, and yield; (ii) grape and wine composition; and (iii) relationships between soil water content and vine, grape, and wine parameters throughout the grapevine growth cycle.

## 2. Materials and Methods

### 2.1. Area of Study

The plots used in the study are in the municipality of Uruñuela, belonging to the DOCa Rioja, and more specifically to the Rioja Alta subzone. The description of the geomorphology and soils of this area is given in [31].

The study area has a Mediterranean climate, according to the UNESCO aridity index, with a slight oceanic influence. Weather conditions throughout the study period were recorded on a daily scale using an agro-climatic station operated by the Government of La Rioja (www.larioja.org/siar, accessed 20 February 2024), situated within the same mesoclimate area (latitude: 42°27′43″ N; longitude: 2°42′46″ W, altitude: 465 m a.s.l.). Figure 1 presents data on annual average temperature, annual average precipitation, annual solar radiation (MJ m$^{-2}$), and annual potential evapotranspiration, as well as precipitation recorded at intervals throughout the vine cycle in the studied years.

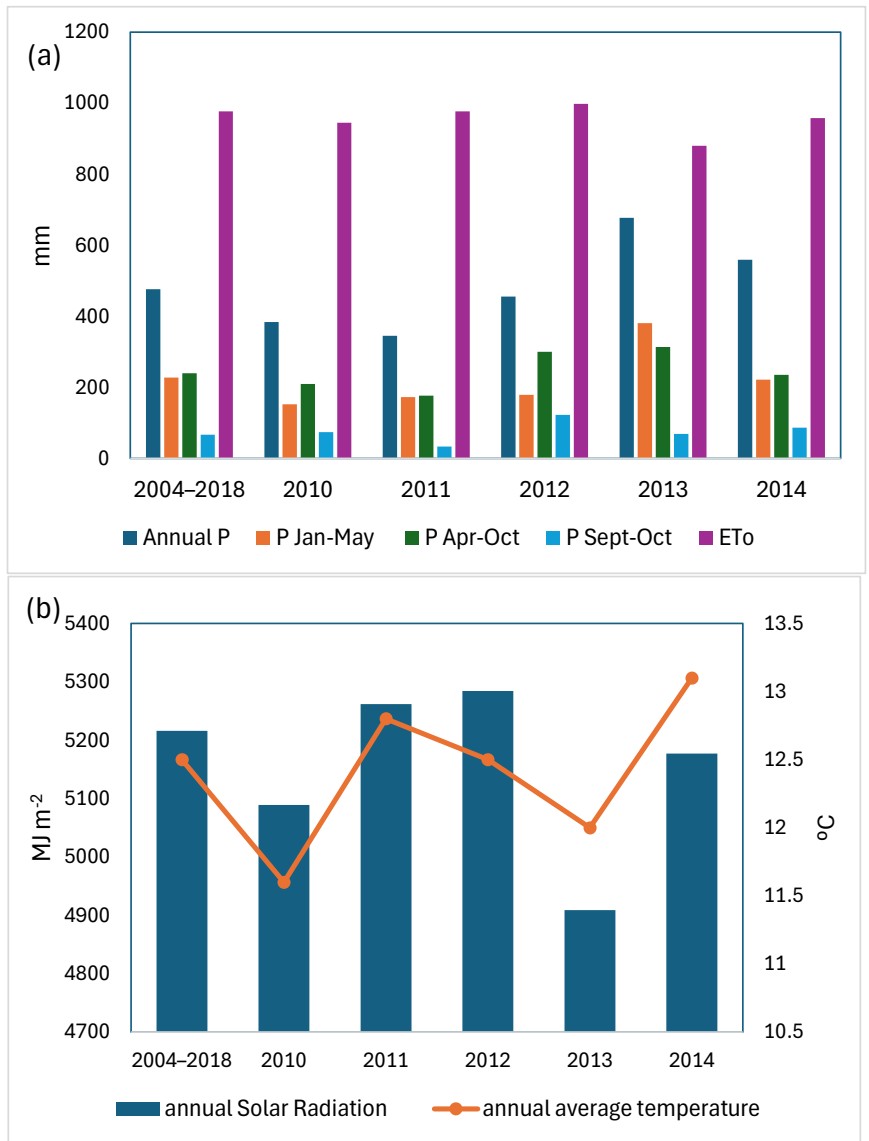

**Figure 1.** Climatic data of years 2010 to 2014 and the historical series 2004 to 2018 (**a**) precipitation and evapotranspiration, and (**b**) annual solar radiation and annual average temperature.

### 2.2. Experimental Design and Vineyard Plot Description

Four commercial vineyards of the Tempranillo variety (*Vitis vinifera* L.) grafted on Richter-110 rootstock were selected for this study. The vineyards were located on platforms with slopes less than 2% (less than one km apart) and were selected for this study. In consequence, climatic conditions can be considered homogenous. The study was conducted over a five-year period (from 2010 to 2014). Within each vineyard, three adjacent rows of 50 vines (plots) were selected. Therefore, each row serves as a repetition for sampling and determinations on vines and berries.

The grapevines were between 20 and 35 years old, with a planting density of 3086 to 3137 grapevines per hectare (Table 1), with vines and row spacing of 1.20 × 2.70 m (vine and row spacing). Double cordon and gobelet were the training systems. Chemical weed control was performed in the row under the vines. In addition, in the interrow, the soil was tilled with a cultivator to eliminate weeds every four to six weeks from February to August.

**Table 1.** Soil characteristics and plant density for each vineyard plot.

| Plot | Plant Density ha$^{-1}$ | Soil Classification (USDA, 2006) | pH (H$_2$O) | E.C. [1] dS/m | O.M. % | Clay % | Silt % | Sand % | CaCO$_3$ % | Available Water Capacity mm |
|------|------|------|------|------|------|------|------|------|------|------|
| | | | | | | Ap Horizon | | | | Section Control |
| P21 | 3086 | Fluventic Haploxerept | 8.15 | 0.15 | 1.00 | 20.6 | 38.8 | 40.6 | 0.5 | 128.5 |
| P22 | 3086 | Fluventic Haploxerept | 8.20 | 0.13 | 1.05 | 24.1 | 28.7 | 47.2 | 1.3 | 146.5 |
| P53 | 3137 | Typic Calcixerept | 8.35 | 0.14 | 0.97 | 18.5 | 52.5 | 29.0 | 3.5 | 56.3 |
| P63 | 3086 | Petrocalcic Palexerolls | 8.40 | 0.15 | 1.87 | 22.9 | 40.0 | 37.1 | 14.7 | 59.9 |

[1] E.C.: electric conductivity; O.M.: organic matter.

### 2.3. Soil Description and Soil Analysis

On 24 May 2010, two soil pits were excavated in each plot for the purpose of describing and sampling the soil horizons. A composite sample of 3 sub-samples (approximately 2 kg) was sampled from each horizon. The soil samples were subsequently air-dried, sieved at 2 mm, and subjected to various analyses. The analyses included pH and electrical conductivity in water (at a ratio of 1:5 soil/solution), organic matter determined by dichromate oxidation [32], soil texture assessed by laser diffraction particle size analyzer (Diffractometer LS$^{TM}$ 13 320, Beckman Coulter, Brea, CA, USA), total carbonate measured by infrared (EQUILAB CO-202, Equilab, Jakarta, Indonesia), active lime content determined through volumetric analysis (Bernard Calcimeter), cation exchange capacity (CEC) using the cobaltihexamine method [33], and extractable potassium analyzed via the Mehlich 3 method [34].

The equations formulated by Saxton and Rawls [35] were used to calculate the soil water holding capacity on each horizon. This process involved incorporating parameters such as electrical conductivity, organic matter content, particle size distribution, and the volume percentage of coarse elements into the calculations. The collective soil water holding capacity for each soil was determined by aggregating the water holding capacities of individual horizons.

Table 1 presents the soil classification and main soil physicochemical characteristics of each plot.

### 2.4. Calculation of Available Soil Water

Soil water was simulated for each plot and year, considering the respective soil properties and weather conditions recorded in each year. The AWC was simulated for each plot based on the climatic conditions in each study year. The simulation was conducted using the Vineyard-Soil Irrigation Model (VSIM—https://www.pvts.net/pdfs/VSIM_Models1/VSIM_5_03.PDF, accessed 20 February 2024). The model performs a daily water balance to determine the water content of the entire soil profile from water inputs, crop evapotranspiration, and drainage. The model also considers the spacing between vines and rows and the cover crop. In a vineyard in the same area, soil moisture data were determined using TDT (Time Domain Transmissometry) GroGraph Moisture Solution probes (ESI Environmental Sensors Inc., Sidney, BC, Canada) at depths of 30, 60, and 100 cm. With this data, the model was calibrated and validated as described in [36].

Following the validation and calibration of the model, the available soil water (ASW) difference between field capacity water and water content at the wilting point was assessed throughout the entire growth cycle for each plot and each year.

According to the Baggiolini scale [37], phenological stages were determined through observations in the experimental plots.

Water stress conditions in grapevines were analyzed by considering the percentage of ASW relative to AWC. According to Pellegrino et al. [38] and van Leeuwen et al. [20], water deficit levels (weak, moderate, and severe) were established with thresholds in terms of the percentage of the AWC as follows: weak water deficit: 32–20% of AWC; weak to moderate

water deficit: 20–8% of AWC; moderate to severe: 8–2% of AWC; and severe water deficit for values <2% of AWC.

### 2.5. Grapevine Nutritional Status

In each row of each plot, 60 leaves were sampled from leaves opposite the second bunch at the veraison stage [39]. Leaf blades and petioles were separated from each leaf, washed with tap water, and rinsed with distilled water. The plant materials were dried at 60 °C in a forced-air oven for 72 h. Then, they were ground through a 0.5 mm sieve using an ultracentrifugal mill (ZM1 Retsch, Haan, Germany).

Nitrogen concentration in petioles (% N) was determined with a CNS elemental analyzer (TruSpec CN, LECO, St. Joseph, MI, USA). Phosphorus (% P) and K (% K) concentrations in petioles were determined by microwave hydrogen peroxide digestion and ICP–optical emission spectroscopy (ICP-3300 DV, PerkinElmer, Shelton, CT, USA). Concentrations were expressed on a dry weight basis as g $100 \text{ g}^{-1}$.

### 2.6. Grapevine Agronomic Performance

All plots were harvested when the grapes had reached 13% $v/v$ probable alcoholic degree, which is the technical maturity for cv. Tempranillo grapes in the DOCa Rioja.

Harvest dates had an interval of 7 to 10 days. At harvest, in each plot, the number of bunches per vine and yield were recorded to calculate grape yield as kg grape $\text{vine}^{-1}$ (GrY) and the average bunch weight as kg (BuW).

At postharvest (late November or early December), 20 vines were randomly selected in each row of each plot, and weight of the pruning wood and the number of shoots were determined to calculate the average shoot weight as g (ShW) and the average pruning wood per vine as kg (PrW).

### 2.7. Grape Sampling and Analytical Parameters of Must

One day before harvest, random samples consisting of 400 berries were collected from six clusters of 20 grapevines randomly dispersed in each row of each experimental plot. Five berries were selected from each cluster, including two from opposite sides at the top of the cluster, two from opposite sides in the middle, and one from the tip of the cluster. In the laboratory, 200 berries were meticulously separated, counted, and weighed to calculate the average weight of each individual berry. Subsequently, a masticator (IUL Instruments GmbH, Königswinter, Germany) was used to crush the berries and extract the must. Probable volumetric alcoholic degree (PVAD), pH (pHM), total acidity (AcTM), malic acid (AcM), and must potassium (KM) were determined using methods outlined by the International Organization of Vine and Wine (OIV) [40]. The total polyphenol index (TPIM) was determined through spectroscopy, measuring the absorbance at 280 nm with a UV–visible spectrophotometer PU8720 (Philips, Eindhoven, The Netherlands). The sample was previously diluted with distilled water to obtain an absorbance within the range. The sample absorbance multiplied by the dilution corresponds to TPIM [41].

An additional set of 200 berries was treated with 1% HCl by heating at 40 °C for 30 min. From this extract, the anthocyanin content in the skin of the berries (AntM) was determined following the method proposed by Ribéreau-Gayon and Stonestreet [42]. Finally, color intensity (CIM) was determined in accordance with the European Community Official Methods. For this purpose, the absorbances at 410 nm, 520 nm, and 620 nm were measured using a spectrophotometer (PU8720, Philips, Eindhoven, The Netherlands). The color intensity corresponds to the sum of these absorbances multiplied by ten [42].

### 2.8. Vinification

Small-scale winemaking was carried out identically for all plots. At harvest, 30 to 45 kg of fruit were taken from each row of each plot. First, the harvested grapes per plot were weighed, crushed, and the stems removed. Subsequently, 3.5 kg of crushed material from each plot were weighed and placed in a glass container adapted for vinification.

Alcoholic fermentation was carried out using a selected yeast strain of *Saccharomyces cerevisiae* (VRB® Uvaferm SIV-Lallemand, Blagnac, France). The fermentation proceeded in a temperature-controlled chamber at 26 °C for 15 days. The final alcoholic grade was approximately 13%. Winemaking was carried out following the protocol described by Sampaio et al. [43].

### 2.9. Wine Analysis

Oenological parameters such as alcohol grade (AlG), pH (pHW), total acidity (AcTW), K (KW), tonality (TnW), and color intensity (CIW) were determined according to OIV methods [44]. Anthocyanin content (AntW) was determined by the Ribéreau-Gayon and Stonestreet method [41], and the total polyphenol index (TPIW) was determined by measuring the absorbance at 280 nm.

### 2.10. Statistical Analysis

The plot factor was selected as the fixed factor and the year factor as the random factor. An ANOVA analysis was conducted with these two factors. When one factor is fixed and the other is random, this type is called a mixed-effects ANOVA. In addition, to determine pair-wise differences by post hoc tests (least significant differences), a one-way ANOVA was also performed.

Pearson correlations were calculated between grapevine, must, and wine parameters with mean ASW in the budbreak–bloom, bloom–fruit set, fruit set–veraison, and veraison–maturity periods. Partial Least Squares regression (PLS regression) analysis between the grape parameters and ASW from one week before stage I2 (week 1) to one week before stage N (week 15) was performed, considering all plots and years.

All statistical procedures were conducted using the STATGPHICS Centurion XV Version 16.1.03 software (Statgraphics Technologies, Inc., The Plains, VA, USA).

## 3. Results

### 3.1. Weather Conditions Recorded during the Period of the Study

The maximum, minimum, mean temperature (TmaxGS, TminGS, and TmGS) and precipitation (PGS) recorded during the growing season (April–October) at the weather station for years from 2010 to 2014 are presented in Table 1. In 2013, precipitation levels were above the average recorded between 2004 and 2018. Although total precipitation in 2012 was below average, autumn precipitation levels exceeded the typical monthly averages. However, the total precipitation for 2012 did not surpass the average of the period considered (2004–2018).

### 3.2. Simulated Soil Water Contents and Available Soil Water for the Selected Plots and Years

Due to the differences in soil characteristics, the maximum AWC that each soil can retain in the profile ranged from 146.5 mm in P22 to 56.3 mm in P63. In the remaining plots, the values were 128.5 and 60.2 mm, respectively, for P53 and P21 (Table 1).

Figure 2 shows the ASW throughout the crop cycle for each plot in the analyzed years, and Figure 3 shows the % of AWC throughout the crop cycle for each plot.

It can be seen that, at the beginning of the crop cycle (in phase D, week ending 25 April), the ASW reached its maximum capacity in almost all plots and years. However, two weeks before the I2 stage (bloom), the ASW decreased rapidly, reaching 50% of the maximum holding capacity within a few weeks, although this varied among plots and years (Figures 2 and 3).

In 2010, plots P21 and P22 reached 50% of their maximum AWC six weeks after bloom, while P53 or P63 reached this level three weeks after bloom. The minimum percentage of the AWC recorded in the vegetative cycle for plots P21 and P22 was 29% and 31%, respectively, reached three weeks after the M1 stage (veraison). For plots P53 and P63, ASW decreased below 20% of AWC (considered as the threshold to define a moderate to weak water deficit) four weeks after the J stage (fruit set). Additionally, in both cases, lower

percentages of the AWC (a minimum value of 11 and 14%, respectively) were reached one week before veraison. Finally, ASW for plots P53 and P63 increased, reaching about 20% of AWC four weeks after veraison (Figures 2 and 3).

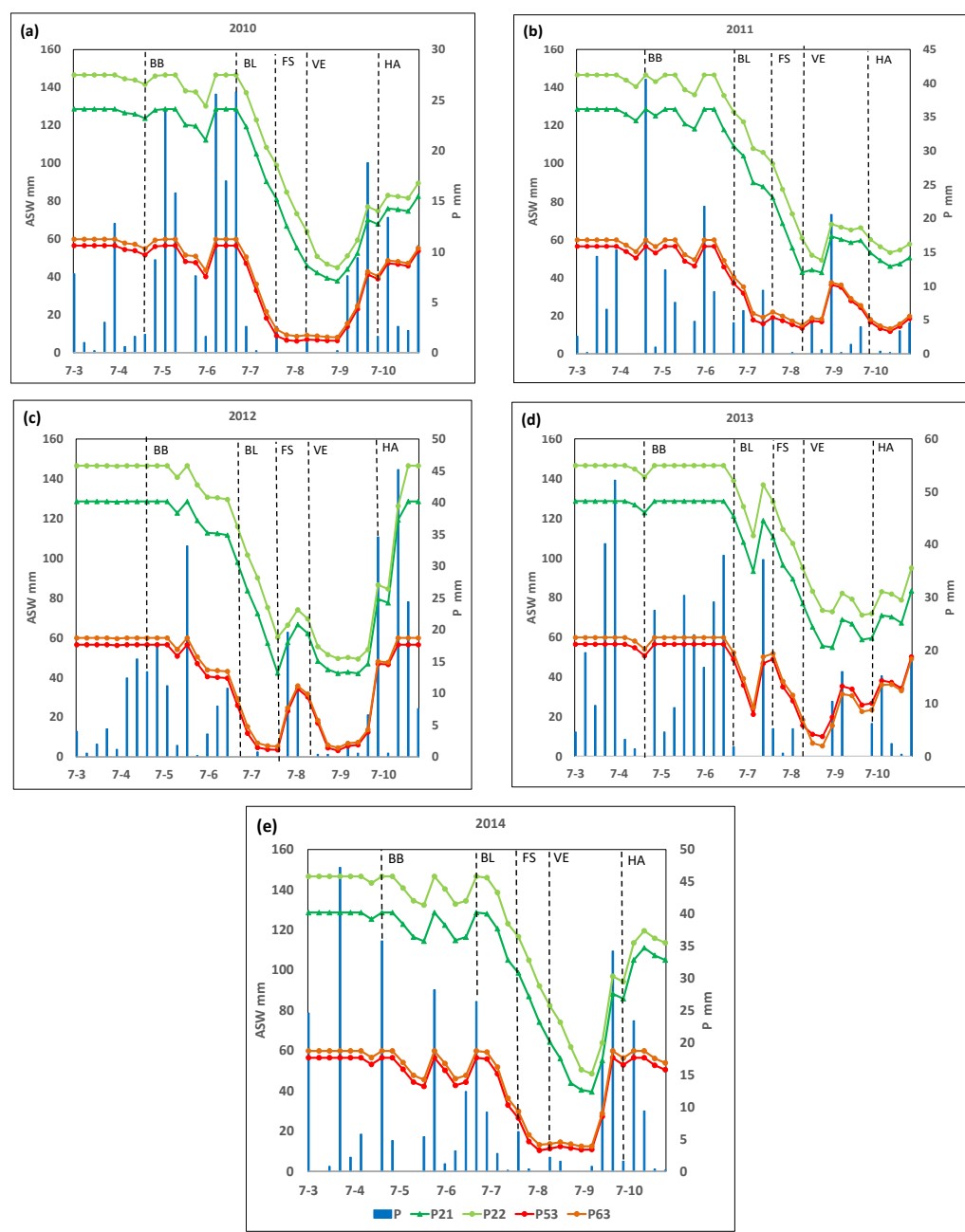

**Figure 2.** Available soil water (ASW, mm) and rainfall (P, mm) along the growing cycle in the selected plots P2, P22, P53, and P63 in years (**a**) 2010, (**b**) 2011, (**c**) 2012, (**d**) 2013, and (**e**) 2014. BB, budbreak; BL, bloom; FS, fruit set; VE, veraison; HA, harvest.

In 2011, P21 and P22 reached 50% of the AWC six and seven weeks, respectively, after the bloom, while plots P53 and P63 reached this threshold two weeks later. In plot P21, the minimum percentage of the AWC about 33%) was reached in veraison, and in plot P22, this minimum percentage of AWC (about 34%) was reached two weeks after veraison, while the minimum value reached in the other two plots (P53 and P63) was about 24 and 25% of the AWC, respectively, and was reached in veraison (Figures 2 and 3).

In 2012, 50% of the AWC was reached three weeks after bloom in P21 and P22, while it occurred at bloom in P53 and P63. The minimum percentages of the AWC in P21 and P22

were 33% and 34%, respectively, and were reached three weeks after veraison. In P53 and P63, 20% of the AWC was reached two weeks after bloom, and the minimum (6 and 9%, respectively) was reached in fruit set. Plots P53 and P63 reached a level above 20% AWC again for four weeks after fruit set. After that, they decreased again to below 20% of AWC until the week of harvest (Figures 2 and 3).

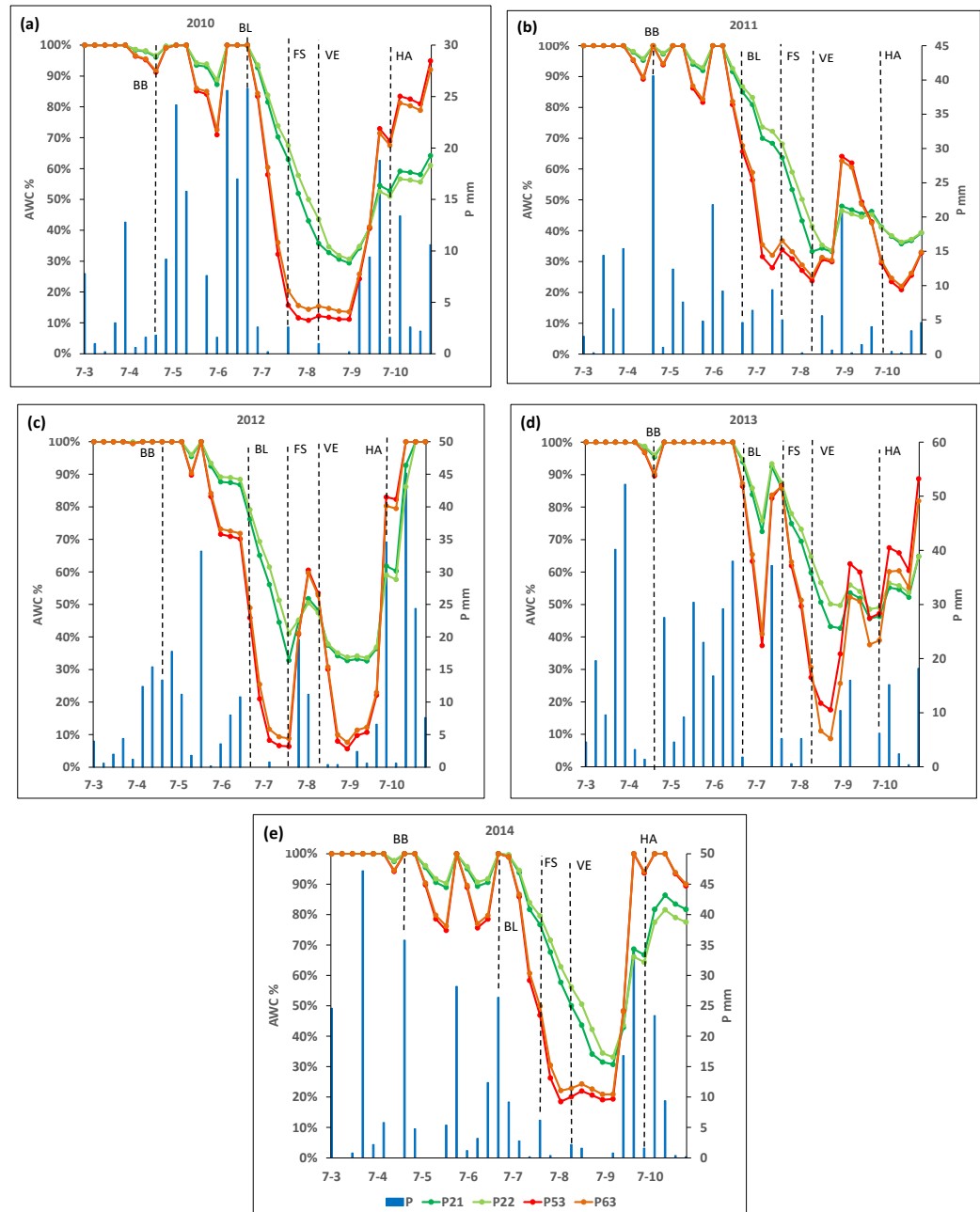

**Figure 3.** Percentage of available water holding capacity (AWC, mm) and rainfall (P, mm) along the growing cycle in the selected plots P21, P22, P53, and P63, in years (**a**) 2010, (**b**) 2011, (**c**) 2012, (**d**) 2013 and (**e**) 2014. BB, budbreak; BL, bloom; FS, fruit set; VE, veraison; HA, harvest.

In 2013, 50% of the AWC was reached two weeks after veraison in P21 and P22, while in P53 and P63, that threshold was reached one week before the veraison. The lowest percentage of the AWC in plots P53 and P63 (18% and 9% of the AWC, respectively) were reached two weeks after veraison. These plots, P53 and P63, recovered values above 20% of the AWC three weeks after veraison (Figures 2 and 3).

Finally, in 2014, both P21 and P22 reached 50% of the AWC one week after veraison, while for P53 and P63, this threshold was reached two and three weeks after bloom, respectively, and one week before fruit set. The minimum percentage of the AWC in P21 and P22 (31% and 33%, respectively) was reached four weeks after veraison, while in plots P53 and P63, the minimum values were 21% and 22% of the AWC, respectively, reached one week before veraison (Figures 2 and 3).

### 3.3. Nutritional Status, Vigor, and Yield

The average values for each plot and year analyzed and the ANOVA pertaining to the nutritional status and vine response are presented in Table 2.

**Table 2.** Comparison of means for foliar contents and vegetative grapevine parameters in each plot (P21, P22, P53, and P63) and year (2010–2014). Results of a mixed-effects ANOVA (plot as a fixed factor and year as a random factor).

| | Petiole % N | Petiole % P | Petiole % K | Shoot Weight g | 100 Berries Weight g | Grape Weight per Vine kg vine$^{-1}$ | Pruning Wood per Vine kg vine$^{-1}$ |
|---|---|---|---|---|---|---|---|
| | | | | 2010 | | | |
| P21 | 0.59b [1] | 0.29c | 2.30c | 161.77b | 244b | 5.54c | 1.55b |
| P22 | 0.56b | 0.28c | 1.23b | 133.89b | 222ab | 4.30ab | 1.32b |
| P53 | 0.45a | 0.18b | 0.78a | 71.17a | 213a | 4.69bc | 0.71a |
| P63 | 0.45a | 0.05a | 0.81a | 75.85a | 208a | 3.55a | 0.71a |
| | | | | 2011 | | | |
| P21 | 0.63c | 0.24c | 2.50c | 183.41c | 312b | 6.10c | 1.75c |
| P22 | 0.56b | 0.28c | 1.18b | 121.83b | 266a | 4.97b | 1.22b |
| P53 | 0.47a | 0.17b | 0.81a | 76.55a | 277ab | 3.42a | 0.75a |
| P63 | 0.45a | 0.05a | 0.78a | 70.09a | 245a | 2.89a | 0.67a |
| | | | | 2012 | | | |
| P21 | 0.55b | 0.33c | 2.64c | 158.59c | 314d | 7.81c | 1.47c |
| P22 | 0.48a | 0.31bc | 1.66b | 129.37b | 264c | 5.69b | 1.17b |
| P53 | 0.45a | 0.22b | 1.11a | 59.50a | 183b | 2.84a | 0.59a |
| P63 | 0.49a | 0.09a | 1.11a | 48.91a | 147a | 2.02a | 0.46a |
| | | | | 2013 | | | |
| P21 | 0.67b | 0.50c | 2.72c | 186.24b | 270a | 3.31a | 1.70c |
| P22 | 0.61b | 0.45c | 1.04b | 157.20b | 258a | 3.87a | 1.34b |
| P53 | 0.45a | 0.19b | 0.71ab | 78.46a | 269a | 4.45a | 0.80a |
| P63 | 0.41a | 0.05a | 0.47a | 74.84a | 245a | 3.29a | 0.69a |
| | | | | 2014 | | | |
| P21 | 0.65c | 0.19b | 2.28b | 142.60c | 339c | 7.04b | 1.42c |
| P22 | 0.48b | 0.28c | 1.09a | 83.42b | 303b | 7.09b | 0.88b |
| P53 | 0.42ab | 0.25bc | 0.87a | 60.34a | 301b | 3.64a | 0.68a |
| P63 | 0.38a | 0.06a | 0.58a | 57.92a | 281a | 3.81a | 0.67a |
| | | | | Average value per year | | | |
| 2010 | 0.51 | 0.20 | 1.28a | 110.70bc | 221.58a | 4.52 | 1.08b |
| 2011 | 0.52 | 0.18 | 1.32a | 112.97bc | 275.14b | 4.34 | 1.10b |
| 2012 | 0.49 | 0.24 | 1.63b | 99.09ab | 226.98a | 4.59 | 0.92a |
| 2013 | 0.53 | 0.30 | 1.24a | 124.18c | 260.58b | 3.73 | 1.13b |
| 2014 | 0.48 | 0.19 | 1.20a | 86.07a | 305.81c | 5.39 | 0.91a |
| | | | | Average value per plot | | | |
| P21 | 0.62c | 0.31c | 2.49c | 166.52c | 295.83b | 5.96c | 1.58c |
| P22 | 0.54b | 0.32c | 1.24b | 125.14b | 262.67ab | 5.18bc | 1.19b |
| P53 | 0.45a | 0.20b | 0.86a | 69.21a | 248.45a | 3.81ab | 0.71a |
| P63 | 0.44a | 0.06a | 0.75a | 65.52a | 225.11a | 3.11a | 0.64a |
| | | | | *p* values | | | |
| Year | 0.4088 | 0.1436 | 0.0099 | 0.0069 | 0.0063 | 0.4377 | 0.0221 |
| Soil | 0.0001 | 0.0001 | 0.0000 | 0.0000 | 0.0129 | 0.0100 | 0.0000 |
| Soil × Year | 0.0007 | 0.0000 | 0.3596 | 0.1850 | 0.000 | 0.0000 | 0.2815 |

[1] Different letters indicate significant differences within soil plots (L.S.D., >95%).

Of all the vine parameters studied, only % K, ShW, BW, and PrW were affected by the year factor. Berry weight was lower in 2010 and 2012 compared to the other years. Pruning weight was also lower in 2012 and 2014 compared to other years. Finally, the % K in the petiole was higher in 2012 than in the other years.

As for the plot factor, this affected all the vine parameters analyzed. So, plots P21 and P22 had higher N, P, and K content, ShW, and PrW than plots P53 and P63, while plot P21 had higher BW, BuW, and GrY than plots P53 and P63.

The interaction between the year and plot was significant for all vine parameters except K content, ShW, and PrW. The significant differences between plots for each year are shown in Table 3 and are not described in detail to avoid excessive length of the text.

**Table 3.** Must analysis for each plot (P21, P22, P53, and P63) and year (2010–2014). Results of a mixed-effects ANOVA (plot as a fixed factor and year as a random factor).

| | Probable Volumetric Alcoholic Degree | pH | Total Acidity g L$^{-1}$ | Malic Acid mg L$^{-1}$ | K mg L$^{-1}$ | IPT | Color Intensity | Anthocyanins mg g$^{-1}$ |
|---|---|---|---|---|---|---|---|---|
| | | | | 2010 | | | | |
| P21 | 13.3a [1] | 3.44ab | 6.52c | 3.80d | 1794b | 13.23a | 3.70a | 1.36a |
| P22 | 13.2a | 3.47b | 5.99bc | 3.09c | 1670b | 15.17b | 4.27ab | 1.62ab |
| P53 | 13.3a | 3.36a | 5.87ab | 2.56b | 1376a | 19.01c | 4.86b | 1.88b |
| P63 | 13.2a | 3.38a | 5.36a | 1.99a | 1296a | 18.79c | 4.72b | 1.83b |
| | | | | 2011 | | | | |
| P21 | 12.8ab | 3.47a | 5.85c | 2.74c | 2005b | 13.07a | 3.32a | 1.19a |
| P22 | 12.6a | 3.49a | 4.93b | 2.32bc | 1793a | 12.38a | 2.78a | 1.11a |
| P53 | 13.5b | 3.44a | 5.11b | 2.05b | 1862ab | 18.90b | 4.44b | 1.80b |
| P63 | 13.2ab | 3.58b | 4.24a | 1.53a | 1724a | 19.22b | 4.88b | 1.64b |
| | | | | 2012 | | | | |
| P21 | 12.8a | 3.60a | 5.47c | 2.81c | 2050b | 14.83b | 3.42a | 0.99a |
| P22 | 12.5a | 3.60a | 4.72b | 2.12b | 1837a | 12.82a | 3.30a | 1.17a |
| P53 | 13.6b | 3.53a | 4.78b | 1.71a | 1944ab | 26.36d | 7.08b | 2.08b |
| P63 | 13.6b | 3.75b | 4.04a | 1.54a | 1946ab | 22.66c | 6.27b | 2.47c |
| | | | | 2013 | | | | |
| P21 | 12.0a | 3.32a | 9.04b | 5.01b | 2038c | 17.32a | 5.70a | 1.28a |
| P22 | 11.8a | 3.28a | 8.48b | 4.49b | 1824b | 16.15a | 6.33a | 1.61b |
| P53 | 13.1b | 3.21a | 7.11a | 3.25a | 1474a | 23.45b | 10.32b | 2.08c |
| P63 | 13.2b | 3.26a | 6.50a | 2.65a | 1373a | 23.76b | 9.93b | 1.64b |
| | | | | 2014 | | | | |
| P21 | 12.2b | 3.41b | 5.90b | 2.90c | 1858c | 12.67a | 3.74a | 1.23a |
| P22 | 11.4a | 3.28a | 5.69b | 2.39bc | 1565b | 12.33a | 3.68a | 1.32a |
| P53 | 14.3c | 3.41b | 5.36ab | 2.22b | 1663b | 18.33b | 5.92b | 1.98b |
| P63 | 13.8c | 3.38b | 4.85a | 1.63a | 1400a | 17.67b | 5.56b | 2.05b |
| | | | | Mean value per year | | | | |
| 2010 | 13.23 | 3.41bc | 5.94c | 2.86b | 1533.75a | 16.55ab | 4.39a | 1.67 |
| 2011 | 13.02 | 3.49c | 5.03ab | 2.16a | 1845.92b | 15.89[a] | 3.85a | 1.43 |
| 2012 | 13.13 | 3.62d | 4.75a | 2.04a | 1944.42b | 19.16bc | 5.01a | 1.68 |
| 2013 | 12.52 | 3.27a | 7.78d | 3.85c | 1677.25a | 20.17c | 8.07b | 1.65 |
| 2014 | 12.92 | 3.37b | 5.45bc | 2.28a | 1621.58a | 15.25a | 4.72a | 1.65 |
| | | | | Mean value per plot | | | | |
| P21 | 12.60a | 3.45 | 6.56c | 3.45d | 1949.07c | 14.22a | 3.98a | 1.21a |
| P22 | 12.29a | 3.42 | 5.96b | 2.88c | 1737.80b | 13.77a | 4.07a | 1.37a |
| P53 | 13.57b | 3.39 | 5.65b | 2.36b | 1663.67ab | 20.42b | 6.52b | 1.97b |
| P63 | 13.40b | 3.47 | 5.00a | 1.87a | 1547.80a | 21.21b | 6.27b | 1.93b |
| | | | | p values | | | | |
| Year | 0.3708 | 0.0000 | 0.0000 | 0.0000 | 0.0046 | 0.0037 | 0.0001 | 0.5773 |
| Soil | 0.0046 | 0.2574 | 0.0001 | 0.0000 | 0.0027 | 0.0000 | 0.0003 | 0.0004 |
| Soil × Year | 0.0003 | 0.0011 | 0.0074 | 0.0050 | 0.0000 | 0.0000 | 0.0000 | 0.0000 |

[1] Different letters indicate significant differences within soil plots (L.S.D., >95%).

### 3.4. Must Composition

The mean values of the variables analyzed in the must as well as the ANOVA results are shown in Table 3.

All the must parameters analyzed were affected by the year factor except for the probable volumetric alcoholic degree (PVAD) and AntM. The year 2013 was the year with the lowest pHM, the highest AcTM, and AcM acid.

Regarding the plot factor, it affected all parameters except pH. Plots P22 and P23 exhibited lower PVAD, TPIM, CIM, and AntM and higher AcM compared to plots P53 and P63. Additionally, plot P21 had higher KM and AcTM than plots P53 and P63.

The interaction among plot and year factors also affected all must variables, and the differences between plots for each year are shown in Table 4.

**Table 4.** Wine analysis for each plot (P21, P22, P53, and P63) and year (2010–2014). Results of a mixed-effects ANOVA (plot as a fixed factor and year as a random factor).

| | Probable Alcoholic Degree | pH | K mg L$^{-1}$ | Total Acidity g L$^{-1}$ | Color Intensity | IPT 186 | Anthocyanins mg L$^{-1}$ |
|---|---|---|---|---|---|---|---|
| | | | | 2010 | | | |
| P21 | 13.8a [1] | 3.79a | 1687b | 6.63c | 9.10a | 47.10a | 567a |
| P22 | 13.5a | 3.80a | 1540b | 6.23b | 10.44a | 49.93a | 614a |
| P53 | 13.3a | 3.70a | 1357a | 6.3bc | 15.69b | 65.83b | 836b |
| P63 | 13.1a | 3.73a | 1317a | 5.63a | 16.92b | 68.87b | 796b |
| | | | | 2011 | | | |
| P21 | 13.5a | 3.87bc | 1713c | 6.13c | 7.89a | 51.13a | 596a |
| P22 | 13.5a | 3.88c | 1537b | 5.43a | 7.74a | 50.13a | 587a |
| P53 | 13.4a | 3.74a | 1280a | 5.83b | 13.87b | 68.47b | 845b |
| P63 | 13.5a | 3.79ab | 1377a | 5.2a | 14.63b | 68.77b | 779b |
| | | | | 2012 | | | |
| P21 | 12.7ab | 3.81a | 1773c | 5.97c | 7.88a | 49.27a | 465a |
| P22 | 12.1a | 3.75a | 1540ab | 5.67b | 6.96a | 46.77a | 486a |
| P53 | 13.0b | 3.83a | 1423a | 5.5b | 13.45b | 80.27b | 996b |
| P63 | 13.3b | 4.02b | 1650bc | 4.37a | 15.25b | 78.57b | 1069b |
| | | | | 2013 | | | |
| P21 | 14.0a | 3.68c | 1923c | 8.83c | 6.49a | 63.43b | 447a |
| P22 | 13.7a | 3.60bc | 1603bc | 7.9b | 7.05a | 52.97a | 508a |
| P53 | 13.5a | 3.47ab | 1263b | 7.43b | 11.82b | 60.40b | 699b |
| P63 | 13.8a | 3.43a | 882a | 6.77a | 12.80b | 60.07b | 630b |
| | | | | 2014 | | | |
| P21 | 12.7b | 3.80b | 1522b | 5.32a | 4.35a | 31.92b | 542b |
| P22 | 11.6a | 3.68ab | 1201a | 5.54a | 3.50a | 27.19a | 439a |
| P53 | 14.9d | 3.78ab | 1437b | 5.33a | 8.81b | 50.68d | 778c |
| P63 | 14.3c | 3.61a | 1129a | 5.58a | 7.99b | 44.82c | 614b |
| | | | | Mean values per year | | | |
| 2010 | 13.43 | 3.75bc | 1475.0 | 6.20b | 13.04c | 57.93b | 703.26ab |
| 2011 | 13.49 | 3.82cd | 1476.7 | 5.65a | 11.03bc | 59.62b | 701.79ab |
| 2012 | 12.79 | 3.85d | 1596.7 | 5.37a | 10.88bc | 63.72b | 754.08b |
| 2013 | 13.75 | 3.54a | 1418.0 | 7.73c | 9.54b | 59.21b | 571.08a |
| 2014 | 13.39 | 3.72b | 1322.3 | 5.42a | 6.16a | 38.67a | 593.17a |
| | | | | Mean values per plot | | | |
| P21 | 13.32 | 3.79 | 1723.7b | 6.58b | 7.14a | 48.57a | 523.47[a] |
| P22 | 12.90 | 3.74 | 1484.2ab | 6.15b | 7.14a | 45.40a | 526.65a |
| P53 | 13.63 | 3.70 | 1371.4a | 6.08b | 12.73b | 64.22b | 830.93b |
| P63 | 13.62 | 3.72 | 1251.6a | 5.51a | 13.52b | 65.13b | 777.63b |
| | | | | *p* values | | | |
| Year | 0.4655 | 0.0037 | 0.3180 | 0.0000 | 0.0000 | 0.0012 | 0.1012 |
| Soil | 0.3728 | 0.4786 | 0.0066 | 0.0095 | 0.0000 | 0.0005 | 0.0004 |
| Soil× Year | 0.0000 | 0.0001 | 0.0000 | 0.0000 | 0.2652 | 0.0000 | 0.0000 |

[1] Different letters indicate significant differences within soil plots (L.S.D., >95%).

### 3.5. Wine Composition

The mean values of the variables analyzed in the wine as well as the ANOVA results are shown in Table 4.

The year factor significantly affected the following wine parameters: pH, AcTW, ICW, TnW, and IPTW. The year 2013 had the lowest pH and the highest total acidity compared to the other years. Additionally, in 2014, the lowest CIW and IPTW were observed, while the highest TnW was recorded compared to the other years.

The soil factor also had a significant effect on KW, AcTW, CIW, TnW, IPTW, and anthocyanin content. Plots P21 and P22 had lower CIW, IPT, and AntW than plots P53 and P63, while plot P21 had higher CIW and KW than plots P53 and P63.

All wine parameters analyzed except CIW had interactions among year and plot factors, the differences between plots for each year are shown in Table 4.

### 3.6. Relationship between Mean Available Soil Water and Grapevine, Must, and Wine Parameters

Regarding the parameters analyzed in grapevine petiole, N, P, and K contents were significantly correlated with the mean ASW in every period studied (Table 5). The ShW and the PrW were correlated with the average ASW in the four periods studied. On the other hand, BW was correlated with mean ASW for periods bloom–fruit set and veraison–maturity, and BuW was correlated with mean ASW for periods budbreak–bloom, bloom–fruit set, and veraison–maturity. However, GrW was correlated with ASW for all four periods studied along the growth cycle.

**Table 5.** Pearson correlation coefficients of foliar contents and grapevine parameters with available soil water (ASW) in different periods of the growth cycle.

|  | Mean ASW mm in Period | | | |
|---|---|---|---|---|
|  | Budbreak–Bloom | Bloom–Fruit Set | Fruit Set–Veraison | Veraison–Maturity |
| Petiole N % | 0.7695 | 0.7282 | 0.7605 | 0.7508 |
| *p* value | 0.0001 | 0.0003 | 0.0001 | 0.0001 |
| Petiole P % | 0.7401 | 0.6896 | 0.7603 | 0.773 |
| *p* value | 0.0002 | 0.0008 | 0.0001 | 0.0001 |
| Petiole K % | 0.6191 | 0.5174 | 0.5261 | 0.5652 |
| *p* value | 0.0036 | 0.0195 | 0.0172 | 0.0094 |
| Shoot weight | 0.8242 | 0.7586 | 0.7618 | 0.7851 |
| *p* value | 0 | 0.0001 | 0.0001 | 0 |
| Berry weight | 0.4139 | 0.499 | 0.4214 | 0.4944 |
| *p* value | 0.0696 | 0.0251 | 0.0643 | 0.0267 |
| Bunch weight | 0.5753 | 0.5515 | 0.4427 | 0.5301 |
| *p* value | 0.008 | 0.0117 | 0.0506 | 0.0162 |
| Grape yield | 0.6273 | 0.6001 | 0.5044 | 0.5893 |
| *p* value | 0.0031 | 0.0052 | 0.0233 | 0.0063 |
| Pruning weight | 0.8306 | 0.7834 | 0.76 | 0.7872 |
| *p* value | 0 | 0 | 0.0001 | 0 |

With respect to the must parameters, PVAD was negatively correlated with the mean ASW for all four periods studied (Table 6). Must pH was not correlated with the mean ASW in any of the periods studied. Total acidity in the must was correlated with ASW in the budbreak–bloom, bloom–fruit set, and fruit set–veraison periods, with a higher r value for the bloom–fruit set, fruit set–veraison and veraison-maturity periods. Similarly, AcM was correlated with mean ASW for all four periods studied with the highest r value in the fruit set–veraison period, while KM was not correlated with mean ASW in any of the periods studied. With respect to TPIM and AntM, there was a negative correlation with the mean ASW for each period studied, while CI was negatively correlated with the mean ASW in the budbreak–bloom, bloom–fruit set, and veraison–maturity periods.

**Table 6.** Pearson correlation coefficients of must properties with available soil water (ASW) in different periods of the growth cycle.

|  | Mean ASW mm in Period | | | |
|---|---|---|---|---|
|  | Budbreak–Bloom | Bloom–Fruit Set | Fruit Set–Veraison | Veraison-Maturity |
| Probable volumetric alcoholic grade | −0.7646 | −0.7687 | −0.844 | −0.8254 |
| *p* value | 0.0001 | 0.0001 | 0 | 0 |
| pH | −0.0338 | −0.2645 | −0.2301 | −0.1316 |
| *p* value | 0.8874 | 0.2597 | 0.3291 | 0.5801 |
| Acidity Total | 0.3903 | 0.498 | 0.5642 | 0.4602 |
| *p* value | 0.0889 | 0.0255 | 0.0096 | 0.0412 |

**Table 6.** *Cont.*

| | Mean ASW mm in Period | | | |
| | Budbreak–Bloom | Bloom–Fruit Set | Fruit Set–Veraison | Veraison-Maturity |
|---|---|---|---|---|
| Malic acid | 0.5693 | 0.6288 | 0.6801 | 0.6037 |
| *p* value | 0.0088 | 0.003 | 0.001 | 0.0048 |
| K | 0.4214 | 0.2451 | 0.3798 | 0.4138 |
| *p* value | 0.0643 | 0.2976 | 0.0986 | 0.0697 |
| TPI | −0.8148 | −0.8111 | −0.6739 | −0.801 |
| *p* value | 0 | 0 | 0.0011 | 0 |
| Color intensity | −0.5543 | −0.4638 | −0.3294 | −0.5098 |
| *p* value | 0.0112 | 0.0394 | 0.1561 | 0.0217 |
| Anthocyanins | −0.805 | −0.7325 | −0.6883 | −0.7576 |
| *p* value | 0 | 0.0002 | 0.0008 | 0.0001 |

In the wine, AlG, pHW, and AcTW were not correlated with the mean ASW in any period studied (Table 7). Potassium in wine was correlated with mean ASW in the budbreak–bloom and veraison–maturity periods. The TnW was not correlated with the mean ASW for any period studied, while TPIW, AntW, and CIW were negatively correlated with the mean ASW in each period studied.

**Table 7.** Correlation coefficients of wine properties with available soil water (ASW) in different periods of the growth cycle.

| | Mean ASW mm in Period | | | |
| | Budbreak–Bloom | Bloom–Fruit Set | Fruit Set–Veraison | Veraison–Maturity |
|---|---|---|---|---|
| Alcoholic degree | −0.365 | −0.2455 | −0.2869 | −0.3281 |
| *p* value | 0.1136 | 0.2968 | 0.22 | 0.1578 |
| pH | 0.1435 | −0.0094 | −0.0282 | 0.0674 |
| *p* value | 0.5462 | 0.9687 | 0.9062 | 0.7776 |
| Acidity total | 0.4061 | 0.4311 | 0.2882 | 0.3287 |
| *p* value | 0.0757 | 0.0577 | 0.2178 | 0.1571 |
| K | 0.5328 | 0.3928 | 0.4397 | 0.4977 |
| *p* value | 0.0156 | 0.0867 | 0.0524 | 0.0256 |
| Color intensity | −0.7649 | −0.8197 | −0.8071 | −0.8302 |
| *p* value | 0.0001 | 0 | 0 | 0 |
| Tonality | 0.3125 | 0.3651 | 0.3606 | 0.4117 |
| *p* value | 0.1797 | 0.1135 | 0.1183 | 0.0713 |
| IPT | −0.6578 | −0.7649 | −0.6381 | −0.689 |
| *p* value | 0.0016 | 0.0001 | 0.0025 | 0.0008 |
| Antochyanins | −0.8025 | −0.8387 | −0.7962 | −0.8244 |
| *p* value | 0 | 0 | 0 | 0 |

*3.7. Relationship between Available Soil Water and Grape Composition*

Figure 4 shows the results of the PLS regression analysis, in which the coefficients for each week start one week before the I2 stage and end one week before maturity is reached. The graph represents the number of components that gave the best fit and lower error, with all variables significantly adjusted with a significance level of 95%. For the PVAD, two components were retained, and the percentage of the variance explained was 75.0%.

For the parameters AcTM, AcM, and pHM, three components were retained, and the percentage of variance explained was 67.6%, 65.4%, and 58.7%, respectively. In the variables associated with phenolic compounds, three components were retained, which explained 92.1%, 67.7%, and 86.8% of the variance for TPIM, AntM, and CIM, respectively. Finally, BW was adjusted by retaining three components, which explained 54.9% of the variance.

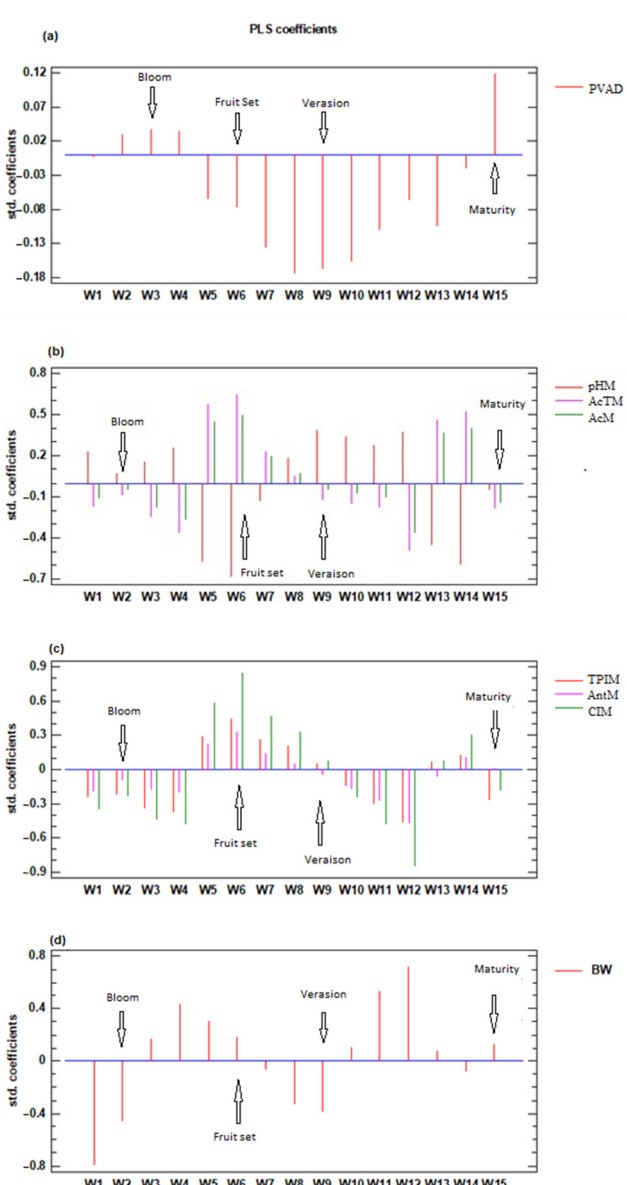

**Figure 4.** Coefficients of the PLS regression analysis were obtained for all analyzed plots and years between the grape parameters and ASW from stage H until reaching ripening (13°) and analyzed weekly (weeks 1–15; W1–W15) (**a**) probable volumetric alcoholic degree (PVAD) (**b**) must pH (pHM), total acidity (AcTM), malic acid (AcM) (**c**) total polyphenol index (TPIM), anthocyanins (AntM), and color index (CIM) (**d**) berry weight (BW).

## 4. Discussion

The properties of the soils in plots P21, P22, P53, and P63 closely resemble those of the vineyard soils in DOCa Rioja, as characterized by Peregrina et al. [45]. The AWC values for these plots fall within the range (30–200 mm) for vine growing soils, as defined by van Leeuwen et al. [46].

Throughout the analyzed period (2010–2014), varying amounts and distributions of rainfall, along with temperature discrepancies, were recorded. These variations provided information on ASW under diverse weather conditions.

It was observed that the soils of P21, P22, P53, and P63 usually maintained the maximum ASW up to bloom (week of 27/6), and from that time, the ASW decreased rapidly. Available soil water levels below 50% of the AWC were reached in all plots during the growing season, usually between bloom and veraison. In a wet year, like 2013, the soil

with higher AWC (P21 and P22) maintained soil water for a longer time. Thus, ASW below 50% of AWC was reached at the beginning of August, five weeks after bloom.

The minimum ASW values were usually reached at veraison, except in very wet years (e.g., 2013). Consequently, in P21 and P22, the minimum ASW values were above 20% of the AWC, while in P53 and P63, the ASW values were below or close to 20% of the AWC between 2 and 6 weeks after bloom. In 2013, this threshold was reached 1 week after veraison. Additionally, minimum ASW values were reached earlier in plots P53 and P63 than in plots P21 and P22, even in the wettest years, such as 2013. Plots P53 and P63 reached values below 10% of the AWC in 2012.

Other authors [36,37,47], reported the evolution of ASW and the minimum values similar to those found in our conditions. These authors expressed ASW in relation to AWC and found values below 0.1 by the end of the growing cycle in some of the years and locations analyzed. The vines experienced weak to moderate water stress at veraison and throughout most of the ripening period, as indicated by the low ASW values. This confirms that the diminished ASW towards the end of the growing cycle not only affected berry weight but also had an impact on berry composition.

Under our conditions, no water stress was observed until the bloom stage. However, moderate water stress (ASW below 20% of AWC) was recorded 2–6 weeks after the I2 stage in plots P53 and P63, with the highest water stress observed at veraison.

The N, P, and K contents in petioles were influenced by the plot factor, with higher contents in plots P21 and P22 than in plots P53 and P63. Additionally, N, P, and K contents were correlated with ASW in all the studied periods. These results could be due to a reduction in nutrient assimilation by mass flow when soil moisture decreases [48], making nutrient absorption by the vine more difficult [9,49]. Furthermore, these three nutrients (as $NO_3^-$, $HPO_4^-$, and $K^+$) are actively taken up by transport proteins against the electrochemical potential with ATP energy input [50]. Thus, a water deficit that promotes stomatal closure and reduced photosynthesis [8], which can affect ATP synthesis, can reduce the uptake of these nutrients. In the case of N, lower ASW would also contribute to a reduction in $NO_3^-$ assimilation, as nitrate reductase activity decreases rapidly and reversibly at low leaf Ψ [51].

The increased N content in plots with higher AWC may contribute to lower concentrations of polyphenols and anthocyanins in musts and wines. This can be attributed to the heightened availability of N, which fosters greater vegetative development in the vines. This vegetative growth competes with the accumulation of sugars and pigments in grapes [52]. Similar findings were reported in other studies under low vine nutritional N status, resulting in elevated anthocyanin and polyphenol content in Cabernet Sauvignon [53] and Merlot [54] berries in the Bordeaux region (France), as well as in Tempranillo grapes in DOCa Rioja [10,55]. The response of the vine to reduced N uptake and lower water availability appears to be additive rather than interactive, as suggested by Keller [8]. In essence, the increased N uptake contributes to a reduced concentration of polyphenols and anthocyanins, aligning with the impact of water stress, as discussed later.

The positive correlations identified between ASW and foliar K content may be attributed to the fundamental role of K in vine water regulation, as noted by Keller [8], given that its assimilation is constrained under limited soil moisture conditions [16,56]. Similar findings were reported by King et al. [57], who observed that vines in areas with higher soil moisture levels exhibited elevated foliar K content. Furthermore, studies conducted in Tempranillo vineyards in La Rioja (Spain) by Zaballa and García-Escudero [58], as well as in Cabernet Sauvignon and Merlot vineyards in Israel by Klein et al. [59], documented a positive relationship between foliar K and increased moisture availability through irrigation.

The plot factor influences ShW and PrW, resulting in lower values observed in plots P53 and P63 compared to plots P21 and P22. This discrepancy may be attributed to the lower ASW after the bloom stage and the moderate water stress experienced in plots P53 and P63. Additionally, the year factor affected ShW and PrW, as evidenced by the highest

shoot weight observed in 2013, which experienced the highest rainfall. Finally, ShW and PrW were found to be correlated with ASW across all periods studied; budbreak–bloom, bloom–fruit set, fruit set–veraison, and veraison–maturity. Our results indicate that shoot growth is highly sensitive to hydric stress, as it primarily involves cell expansion [60,61]. A reduction in shoot and leaf growth is typically the first visible sign of a water deficit in the vineyard [62,63]. Similar results, showing increased vine vigor in soils with higher AWC, have been reported by Tramontini et al. [7] in Bordeaux and by Tomasi et al. [64] in Italy. The weight of the berries exhibited a strong correlation with ASW, indicating higher values in wetter years compared to drier ones [65–68]. Coinciding with those results, in our study, ASW had a significant positive correlation with berry weight in the veraison–maturity period. Additionally, the PLS coefficients were high and positive during the weeks after veraison (W10 to W13). This effect after veraison is consistent with van Leeuwen et al. [20], who found smaller berry sizes under water limitations at ripening. In our conditions, BW showed high and negative PLS coefficients with ASW around bloom (W1 to W2) and veraison (W7 to W9). These results are according to those reported by Ramos et al. [36]. The explanation may be connected to the observation that, during these two periods, the balance between vegetative and reproductive growth becomes more crucial than in other phases. The increased ASW tends to promote vegetative growth over berry development in these specific periods.

Grape yield showed slight differences with BW, as it was affected by the plot factor but not by the year factor. Moreover, production correlated with ASW during the periods from budbreak to flowering and from veraison to ripening. A similar result with yield affected by water available during the budbreak–bloom period was reported by Ramos and Martínez-Casasnovas [15]. The relationship during the veraison to ripening period may be because hydric stress can affect photosynthesis, and lower yields may occur because not all berries reach full maturity [69].

In addition to berry development, a water deficit can also affect grape quality. Thus, PVAD was affected by the plot factor, with higher values in plots P53 and P63 compared to plots P21 and P22. These results would be due to the higher water stress in plots P53 and P63, as confirmed by the correlation between PVAD and ASW in all the periods studied. Moreover, PLS coefficients were negative during most of the period studied but were higher around veraison (W8–W10), showing that this period would be the most influential on the final PVAD. The positive PLS coefficients in the weeks prior to harvest agree with the results reported by Lasko and Pool [70], where water stress in the following weeks to veraison was not as critical as at the end of the cycle, when it can produce both a decrease in berry size and sugar accumulation. Thus, Bucelli et al. [71] and Ubalde et al. [72] reported similar results, indicating higher sugar content in grapes from vineyards with lower AWC soils and shallower soil rooting depth [71].

Finally, the alcoholic content of the wine (AlcG) was not affected by either the plot or the year factors. This would indicate that changes in the PVAD were not significant enough to impact the AlcG of the wines.

Must acidity is another parameter that can be directly affected by soil water availability. In our results, pH, total acidity, and malic acid were influenced by the year factor but not by the plot factor. These results confirm that climate has a greater influence on the acidity components than soil, as observed by van Leeuwen et al. [13].

The year 2013, the wettest of the analyzed years, had a lower pH and higher total acidity and malic acid levels than other years. Additionally, AcTM correlated with ASW during all periods studied except budbreak–bloom, while AcM showed a positive correlation in all the periods studied. Consistent with our results, higher acidity has been reported in wetter soils [1], in conditions of excessive soil moisture [73], and with well-watered grapes [74,75]. Additionally, Lopes et al. [76] reported a significant reduction in titratable acidity of the must when soil water content was reduced during spring. Likewise, Tomasi et al. [71] found higher acidity in soils with greater rooting depth. Furthermore, studies by Peyrot des Gachons et al. [77], de Souza et al. [78], and dos Santos et al. [79] have documented reduced

total acidity attributed to water deficit and/or alterations in temperature in sun-exposed berries in rainfed vines.

As for the time of the cycle in which some effects on both the total acidity and malic acid in the must, the reasons may vary.

Enhanced water availability in the early phases of fruit growth (W5 to W8), especially when the berry is small and acid synthesis begins, could have a favorable impact on final acidity levels. This correlation corresponds to the positive PLS coefficients observed during this timeframe. Conversely, the negative PLS coefficients in the following period (W9 to W11) may be attributed to a dilution effect as the berry undergoes rapid expansion. Moreover, it was noted that elevated ASW during ripening contributes to heightened acidity levels.

Our result is according to Ramos and Martínez de Toda [67], who, in a comprehensive study covering nearly the entire Rioja vine-growing region, identified the highest acidity values in the wettest years. These years witnessed substantial water accumulation not only from bloom to veraison but also from veraison to maturity. Furthermore, in those instances, the lower temperatures recorded in wetter years could have potentially impeded the combustion of malic acid during the ripening process.

These changes in grape acidity components were reflected in the wine. Thus, the pH and total acidity of the wine were influenced by the year factor and not by the plot factor. Similarly to the must, 2013 exhibited the highest acidity and the lowest pH of the wines. These results confirm the importance of the climatic conditions of the year on the final acidity of the wines. Another must component related to grape quality is K. Under our conditions, must K (KM) was affected by both plot and year factors. Thus, P21 and P22 soils, with higher AWC, had higher KM content. Increased ASW can raise K levels in vegetative tissues (as above mentioned), leading to higher K levels in berries as they serve as significant sinks for K [11]. While for K content in wine, we found results similar to those found in must, indicating the influence of both the year and plot factors. Additionally, correlations between ASW and KW were observed in the budbreak–bloom and veraison-maturity periods. These are important results since K reduces the free acids in the wine and raises its pH, leading to lower chemical/biological stability of the wine.

Grape anthocyanins, total phenol grape content, and color intensity (CIM) were affected by the plot factor. Thus, soils P53 and P63, with lower AWC and experiencing moderate to severe postveraison water stress, showed higher CIM, TPIM, and AntM. Moreover, in all the periods studied (budbreak–bloom, bloom–fruit set, fruit set–veraison and veraison–maturity), TPIM and AntM were negatively correlated with ASW. These results agree with the observations of other authors [36,67,80–82], although different compounds may be affected differently depending on cultivars and rootstocks [80,83]. Anthocyanin and polyphenol concentrations improved in scenarios where there was an absence of water stress from bloom to fruit set, mild water stress between fruit set and veraison, and moderate to severe water stress post-veraison [84]. This agrees with the positive PLS coefficients for CIM, TPIM, and AntM observed in the weeks W5 to W8, and with the negative PLS coefficients observed after veraison (W10–W12). In this regard, Ferrer et al. [82] noted higher anthocyanin concentrations under mild to moderate water deficits during ripening.

The periods following W9, during which a negative relationship between ASW and phenolic compounds was observed, coincide exactly with the periods in which the ASW increases berry size. Therefore, it could be suggested that the higher concentration of phenolic compounds is influenced by ASW and its effect on berry size. Changes in polyphenol and anthocyanin content observed in the must were also found in the wine. Both the plot factor and the year factor affected CIW, TPIW, and AntW. Additionally, ASW was negatively correlated with these wine properties across all of the studied periods. This result is important as it confirms the impact of AWC on wine quality. A similar result was found in Spain by Ubalde et al. [72], with a higher anthocyanin content in wine from soils with lower AWC.

Between stage H and maturity, there were only two short periods lasting two to three weeks each (except for one lasting five weeks in the case of acidity), during which the impact of water availability diverged, albeit with lesser intensity compared to the overall effect throughout the rest of the vegetative period. The elevated ASW during these specific periods led to a reduction in berry weight and acidity while concurrently boosting the levels of anthocyanins and other phenolic compounds.

These two periods consistently align with the bloom and veraison stages. One plausible explanation may be linked to the general physiological, and especially hormonal, changes observed in the vine during these stages. In both instances, vegetative growth decelerates concurrently with reproductive development—in one instance to form the fruit and in the other to commence the ripening phase [85].

## 5. Conclusions

From this research, it could be concluded that the evolution of available soil water and the soil and the level of water stress under similar weather conditions (mesoclimate) varied among the study plots due to different soil characteristics (especially the effective soil depth) that determine different available water holding capacities. These differences in available water influence nutritional status, vegetative development, and grape and wine composition. Higher available soil water between the budbreak and ripening phenological stages increased leaf N, P, and K levels. Vegetative development and grape yield per vine also increased, while the probable alcoholic degree of the berries decreased. Higher available soil water from veraison to harvest stages also reduced the berry color and polyphenol and anthocyanin content of the berries. As for acidity components, higher soil water availability around fruit set increased total acidity and malic acid. The impact of soil available water on berry characteristics was transmitted to the wine in the case of color, polyphenols, and anthocyanins, which decreased in plots with higher available soil water, but not in the case of alcohol content, pH, and total acidity. Therefore, the complete description of the soil profile with its rooting depth and the determination of the available water holding capacities can be useful for the selection of the most favorable soils for obtaining quality grapes in current climatic conditions. In addition, in a scenario of climate change where soil water could decrease due to changes in rainfall and increased demand for evapotranspiration, the location of vineyards in soils with greater available water holding capacities should be considered.

**Author Contributions:** Conceptualization, F.P.; methodology, J.M.M.-V. and M.C.R.; validation, M.C.R.; formal analysis, F.P. and J.M.M.-V.; investigation, F.P., J.M.M.-V. and E.P.P.-Á.; resources, J.M.M.-V.; data curation, J.M.M.-V.; writing—original draft preparation, F.P.; writing—review and editing, F.P., J.M.M.-V. and E.P.P.-Á.; visualization, F.P. and J.M.M.-V.; supervision, E.G.-E.; project administration, J.M.M.-V. and E.G.-E.; funding acquisition, J.M.M.-V. and E.G.-E. All authors have read and agreed to the published version of the manuscript.

**Funding:** This work was supported by MINECO, Instituto Nacional de Investigación y Tecnología Agraria y Alimentaria INIA, European Social Fund [grant number RTA2009-00101-00-00], and the Government of La Rioja. Eva Pilar Pérez-Álvarez also acknowledges the Spanish Ministry of Science, Innovation and Universities (MCIU) for her "Juan de la Cierva Incorporación" postdoctoral grant (IJC2019-040502-I).

**Data Availability Statement:** The original contributions presented in the study are included in the article, further inquiries can be directed to the corresponding author.

**Acknowledgments:** We gratefully acknowledge Zinio Wineries from Uruñuela (La Rioja) for its collaboration with selection of vineyards plots.

**Conflicts of Interest:** The authors declare no conflicts of interest.

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
