# Peer review of "Differences in Soil Water Holding Capacity and Available Soil Water along Growing Cycle Can Explain Differences in Vigour, Yield, and Quality of Must and Wine in the DOCa Rioja"

_horticulturae, doi:10.3390/horticulturae10040320_

Round 1

Reviewer 1 Report

Comments and Suggestions for Authors

The paper analyses the effects of soil water holding capacity and available soil water, during the vegetative cycle, on some vegeto-reproductive characteristics of grapevine variety Tempranillo in the DOCa Rioja, during 4 years.

The original part of the work lies in the duration of the experimentation and its execution in 4 areas of the Rioja denomination. The results are consistent with other similar works and with what is reported in the literature, but the novelty lies in describing the relationships between the water available in different phenological phases of the vine, with reflections also in the wine obtained.

The application of a multivariate analysis, such as PCA, would have better highlighted the effects of the 4 zones and 4 years.

A suggestion: since there are many acronyms, it could be better to indicate more often the entire text of parameters, or provide a table with their synthesis

Paper can be accepted with some revisions (see attached file).

Comments on the Quality of English Language

Moderate English revision is needed

Reviewer 2 Report

Comments and Suggestions for Authors

The manuscript "Differences in soil water holding capacity and available soil water along growing cycle can explain differences in vigour, yield, and quality of must and wine in the DOCa Rioja" was well prepared and developed. Admittedly, the manuscript does not bring new knowledge to the area studied, but only confirms this knowledge. The influence of water and temperature conditions on the content of polyphenolic compounds is quite well described in the literature. The authors characterized the conditions of the experiment in sufficient detail. The text only needs some corrections to clarify what is selected in the uploaded file
